# A Markov chain method for weighting climate model ensembles

Max Kulinich[1], Yanan Fan[2], Spiridon Penev[3], Jason P. Evans[4], and Roman Olson[5]

[1,2,3]School of Mathematics and Statistics, UNSW Sydney, Australia.
[4]Climate Change Research Centre and ARC Centre of Excellence for Climate Extremes, UNSW Sydney, Australia
[5]Irreversible Climate Change Research Center, Yonsei University, South Korea

**Correspondence:** Max Kulinich (m.kulinich@student.unsw.edu.au)

**Abstract.** Climate change is typically modelled using sophisticated mathematical models (climate models) of physical processes that range in temporal and spatial scales. Multi-model ensemble means of climate models show better correlation with the observations than any of the models separately. Currently, an open research question is how climate models can be combined to create an ensemble mean in an optimal way. We present a novel stochastic approach based on Markov chains to estimate model weights in order to obtain ensemble means. The method was compared to existing alternatives by measuring its performance on training and validation data, as well as model-as-truth experiments. The Markov chain method showed improved performance over those methods when measured by the root mean squared error in validation and comparable performance in model-as-truth experiments. The results of this comparative analysis should serve to motivate further studies in applications of Markov chain and other nonlinear methods that address the issues of finding optimal model weight for constructing ensemble means.

## 1 Introduction

Climate change is often modelled using sophisticated mathematical models of physical processes taking place over a range of temporal and spatial scales. These models are inherently limited in their ability to represent all aspects of the modelled physical processes. Simple averages of multi-model ensembles of GCMs (Global Climate Models) often show better correlations with the observations than any of the individual models separately (Kharin and Zweirs (2002); Feng et al. (2011)). Knutti et al. (2010) point out that often the equal-weighted averages ("one model, one vote") approach is used as a best-guess result, assuming that individual model biases will at least partially cancel each other out. This approach assumes that all models are (a) reasonably independent, (b) equally plausible, (c) distributed around reality and (d) that the range of their projections is representative of what we believe is the uncertainty in the projected quantity. However, these assumptions are rarely fulfilled (Knutti et al. (2017)), and thus a better way of finding a weighted ensemble mean is required (Herger et al. (2018); Sanderson et al. (2017)).

Most studies attempting to define an optimal ensemble weighting either employ linear optimisation techniques (Krishnamurti et al. (2000); Majumder et al. (2018); Abramowitz et al. (2018)) or are based on a specification of likelihoods for the model and observation data (Murphy et al. (2004); Fan et al. (2017)). Such methods are inevitably limited by the strong assumptions used for their design. We seek to weaken those assumptions and to complement the existing methods with a more flexible

nonlinear optimisation approach. An unresolved issue in using weights for models is that models have interdependence, due to the sharing of computer codes, parameterizations, etc. (Olson et al. (2019)). Abramowitz et al. (2018) points out that model dependence can play a crucial role when assembling the models into an ensemble. Mathematically, interdependence often result in closeness of model outputs in model output space. If a large cluster of highly dependent models is included into an ensemble with equal weights, the overall ensemble mean will become close to the dependent models' cluster. Ignoring model dependence can lead to bias and overconfidence in future climate model projections (Leduc et al. (2015); Steinschneider et al. (2015)).

Hence, it is desirable that an ensemble weighting method is robust against the dependency issue, and has normalised non-negative weights for interpretability. Finally, the methods should work well across a range of different climate variables, such as temperature, precipitation, etc.

In this paper, we propose a novel way to construct a weighted ensemble mean using Markov chains, which we call the Markov Chain Ensemble (MCE) method. Our purpose is to demonstrate that going beyond linear optimisation on a vector space of climate models' outputs allows building better performing weighted ensembles. We selected Markov chains as a basis for such nonlinear optimisation as one of the most straightforward nonlinear structures. It naturally produces non-negative weights that sum to one and captures some of the nonlinear patterns in the ensemble (here we refer to nonlinear patterns as time-dependent selection of model components rather than considering complete model output vector). It performs well on a range of datasets when compared to the standard simple mean and linear optimisation weighting methods as we demonstrate below. We also examine how the method responds to the introduction of interdependent models.

Although Markov chains have been used frequently in the literature for the prediction of future time series (e.g. Bai and Wang (2011); Pesch et al. (2015)), to the best of our knowledge, this is the first time the method has been applied to building weighted climate model ensemble means. In this paper, we use the "memoryless" property of Markov chains at each time step to capture the dynamic change in models' fit through the time series. This dynamic change, through time, is represented by the transition matrix, which describes the probability of each model being the best fit for the next observation at time $t + 1$, given the best fit for the current time $t$. The transition matrix is built based on the input data and describes probable future states given the current state. The stationary distribution of this transition matrix is used for weighted ensemble creation and reflects the relative contribution of each model to the total weighted ensemble mean forecast.

We describe the datasets used in this study and the proposed MCE method in Section 2. We compare the proposed method (MCE) to the commonly used multi-model ensemble average (AVE) method (Lambert and Boer (2001)) and the convex optimisation (COE) method proposed by Bishop and Abramowitz (2013) and present the results in Section 3, followed by a discussion in Section 4 and conclusion in Section 5.

## 2 Methods

### 2.1 Data

Here we first describe the datasets used in this study. We have chosen three publicly available datasets with differing number of models, historical period lengths and model interdependence levels to evaluate and compare the performance of the MCE method with alternative approaches.

**CMIP5 Data**: The first dataset we use is the temperature anomalies (°C) data from Coupled Model Intercomparison Project (CMIP5) with 39 different Global Climate Model (GCM) outputs (one ensemble member per model) and Hadley Centre/Climatic Research Unit Temperature observations (HadCRUT4). The data is obtained from `https://climexp.knmi.nl` and the period of 1900 - 2099 is selected for the analysis. It contains temperature anomalies (monthly averages) compared to the reference period of 1961-1990 (Taylor et al. (2011)). This dataset contains several clusters of dependent models, has both positive and negative data values, a relatively low variability and long time series.

**NARCliM Data**: The second dataset contains temperature output from the New South Wales (NSW) and Australian Capital Territory Regional Climate Modelling project (Evans et al. (2014)). It contains regional climate model (RCM) simulations over southeastern Australia. Specifically, three RCM versions were forced with four global climate models each, for a total of twelve ensemble members. The data contains annual time series of mean summer temperature (°C) for the Far West NEW state planning region as modelled by the NARCliM domain regional climate models (RCMs) for the periods 1990–2019 and 2030-2039 (Olson et al. (2016)). Corresponding temperature observations are obtained from the Australian Water Availability Project (AWAP) (Jones et al. (2009)). The dataset has a high ratio of the number of models to the number of observations. While NARCliM model choice explicitly considered model dependence for both the RCMs as well as the driving GCMs, the resulting ensemble demonstrates an apparent similarity between the simulations (i.e., model inter-dependence) in small clusters.

**KMA Data**: The third dataset contains yearly heatwave amplitudes (HWA) for the Korean peninsula from 29 CMIP5 climate models and observations between years 1973 and 2005 (Shin et al., 2017). In particular, HWA contains the difference between the highest temperature during the heatwave events for the corresponding year and the 95th percentile of daily maximum summer temperatures from 1973 to 2005. This framework was discussed in detail in Fischer and Schär (2010). Here a heatwave event occurs when the daily maximum temperature is above the 95th percentile of daily maximum summer temperatures (32.82°C) for two consecutive days. Daily maximum temperature data used for the calculation of observed HWA is the mean of 59 weather stations operated by the Korea Meteorological Administration (KMA). Shin et al. (2017) provides the list of CMIP5 models included in the study. HWA data is non-negative and can be highly skewed with long upper tails as it measures extreme events; therefore, the dataset is highly non-Gaussian. These properties allow us to test methods in more challenging scenarios, where likelihood-based approaches are more difficult to apply.

These three datasets cover different scenarios, data structures, parameter distributions and scales (see Table 1) . Such coverage allows us to analyse the performance and the inherent limitations of the proposed method. In this pilot study we use

spatially averaged data, which limits physical interpretability of the model weights, but the method can be extended to spatially distributed data.

| Dataset | Climate variable | Minimum | Maximum | Variance | Number of observations | Number of models |
|---------|------------------|---------|---------|----------|------------------------|------------------|
| CMIP5 | Temperature (°C) | -0.80 | 1.16 | 0.12 | 1440 | 39 |
| NARCLiM | Temperature (°C) | 9.38 | 31.64 | 36.61 | 240 | 12 |
| KMA | HWA (°C) | 0 | 1.49 | 0.18 | 33 | 29 |

**Table 1.** Summary of CMIP5, NARCLiM and KMA data properties.

## 2.2 Markov chain ensemble (MCE) method

Generally, a homogeneous Markov chain is a sequence of random system states evolving through time, where each next state is defined sequentially based on its predecessor and predefined transition probabilities (Del Moral and Penev, 2016, p. 121). Suppose that there is a finite number of probable system states $S = \{s_1, \ldots, s_N\}$, then this dependency can be described through a transition matrix $P$ (with $P(x,y) \in [0,1]$ and $\sum_y P(x,y) = 1$, for any $x, y \in S$):

$$\forall x, y \in S, \quad Pr(X_{n+1} = y | X_n = x) = P(x,y). \tag{1}$$

In this study, we want to utilise the "Fundamental Limit Theorem for Regular Chain" which states that if $P$ is a transition matrix for a regular Markov chain (where $\forall x, y \in S, \ P(x,y) > 0$), then $\lim_{n \to \infty} P^n = P^\infty$ where $P^\infty$ is a matrix with all rows being equal and having strictly positive entries.

This property allows us to construct a non-negative transition matrix $P$ by distillation of input information (i.e., model outputs and historical observations) and allows $P$ to converge to a unique vector of model weights $w = (w_1, w_2, ..., w_N)$, where $N$ is a total number of models in a given ensemble. The vector $w$ can be obtained by solving the equation $wP = w$. The converged transition matrix represents a probability of selecting one of the models for any of the time steps in the future when observations are not available. Hence, we propose to use it as a weighting vector for constructing a weighted ensemble mean forecast and test this proposition using cross-validation in the following sections.

More precisely, we start by constructing a transition vector $v$ (based on the input data) which specifies a choice of the optimal model at any given time step $t$. Using the vector $v$ we construct a transition matrix $P$ and find its stationary distribution $w$. The resulting weighted ensemble mean is constructed by applying $w$ on the given climate model outputs. We call this process Markov Chain Ensemble (MCE) algorithm, and it uses historical observations and equivalent climate model simulations as the input data to calculate a set of weights for the future ensemble mean as an output. Table 2 gives a step by step description of the MCE algorithm.

**Input:**

- length of training period $T_1$, and
- historical observations $O_t$, at times $t = 1, \ldots, T_1$, and
- climate model output $M_{i,t}$, at times $t = 1, \ldots, T_1$, for $i = 1, \ldots, N$ models, and
- an initialised number of simulations $L$
- an initialised $\sigma$ interval $[\sigma_{min}, \sigma_{max}]$
- an initialised transition matrix $P^0$ of $N$ x $N$ size

**Step 1.** Randomly select $\sigma \in [\sigma_{min}, \sigma_{max}]$ and compute the distance matrix $D$ according to Equation 2.

**Step 2.** Construct a sequence vector $v$ based on $D$ using stochastic simulations.

**Step 3.** Update $P^0$ step-wise by increasing probability of transitions contained in $v : P^0 \to P^1 \to \ldots \to P^{T_1}$.

**Step 4.** Obtain normalised transition matrix $P^*$, by normalising $P^{T_1}$ row-wise so that each row sums to 1.

**Step 5.** Find $w$ by solving $wP^* = w$ and store its value.

**Step 6.** Construct the ensemble mean based on weights $w$ and calculate its $RMSE_{Tr}$

**Step 7.** Repeat Step 2 - 6 until $L$ sets of weights $w^1, w^2, \ldots, w^L$ and respective $RMSE_{Tr}^1, RMSE_{Tr}^2, \ldots, RMSE_{Tr}^L$ have been obtained.

**Step 8.** Select a set of weights $w^*$ corresponding to the minimal $RMSE_{Tr}^*$

**Step 9.** Construct the final $E_{MCE}$ using the selected $w^*$

**Table 2.** The Markov Chain Ensemble (MCE) algorithm.

We provide some details of the algorithm as described in Table 2 in the following paragraph.

**Initialisation of transition matrix $P^0$:** In order for Markov chain to be regular we set $P^0(x,y) = \lambda, \forall x, y \in S$, where $\lambda$ equals the lowest computationally possible positive number $\lambda = 2.225074e^{-308}$ in the R software (R Core Team (2013)).

**Initialisation of $\sigma$ interval:** To avoid division by 0 in Equation 2 and to prevent Equation 2 from converging to $1/N$ the initial $\sigma$ interval is set to $[0.1, 1]$.

**Step 1:** The MCE method proceeds by utilising each model output in an optimal way based on its ability to resemble observational data at each given time point. This resemblance is measured by a distance-based probability matrix $D$ of size $N \times T_1$, using a normalised exponential function.

$$d_{i,k} = \frac{e^{-\left(\frac{M_{i,k} - O_k}{\sigma}\right)^2}}{\sum_{j=1}^{N} e^{-\left(\frac{M_{j,k} - O_k}{\sigma}\right)^2}} \tag{2}$$

where $1 \leq k \leq T_1 \leq T$, $T_1$ indicates the length of the training period, and $T$ is the length of the entire historical period included in the study. Additionally, $1 \leq j \leq N$ where $N$ is the number of models included, and $\sigma$ is chosen randomly as described above.

**Step 2:** Based on the matrix $D$ a simulation is performed at each time step $1 \leq k \leq T_1$ by randomly selecting one of the models $i$ with probability proportional to its value $d_{i,k}$. This way we construct a vector $V = (v_1, v_2, v_3, ..., v_{T_1})$, which represents choice of models closest to observations at each time step.

**Step 3:** Then the initial matrix $P^0$ is updated step-wise $(P^1, P^2, ..., P^{T_1})$ to capture the transitions between models present in vector $V$. For each $t$ $(1 \leq t \leq T_1 - 1)$, $P^i_{V_i, V_{i+1}} = P^{i-1}_{V_i, V_{i+1}} + 1$.

**Step 4:** The resulting matrix is normalised by row $P^*_i = P^{T_1}_i / \sum_{j=1}^{N} P^{T_1}_{i,j}$, for each $1 \leq i \leq N$.

**Step 5:** The stationary distribution $w$ is obtained by solving $wP^* = w$. A standard R software package is used to find the solution in this study.

**Step 6:** Construct the ensemble mean based on weights $w$ and calculate its $RMSE_{Tr}$.

**Step 7:** Steps 2 - 6 are repeated $L$ times, where $L$ is selected based on the external requirements on precision of the results and on computational power available.

**Step 8.** Select the set of weights $w^*$ with the best performance on the training set (with the lowest $RMSE^*_{Tr}$).

**Step 9.** Construct the $E_{MCE}$ ensemble using the selected $w^*$.

### 2.2.1 Parameter sensitivity

From Equation 2 it is clear that having a small $\sigma$ will result in distances $d$ close to $1/N$. Having a large $\sigma$ will result in all the distances becoming marginal with the exception of the largest one. To optimize the properties of the simulations we control $\sigma$ by randomly choosing it from $[0.1, 1]$ interval.

As we select only one of the simulations, the MCE method is not sensitive to the number of simulations $L$ after a certain threshold. This threshold is set based on the requirements for precision of the results and on the calculation time. In Figure 1 we illustrate the simulation performance dynamics (simulation index and performance on training and validation NARCLiM data) depending on value of $L \in [1, 1000000]$. The simulation index $i^* \leq L$ represents the index of the best performing simulation at each value of $L$ (with $w^*$ vector of weights and $RMSE^*_{Tr}$ as descibed in Step 8 of Table 2). The cross-validation procedure and RMSE metrics are described below in Sections 2.5 - 2.6.

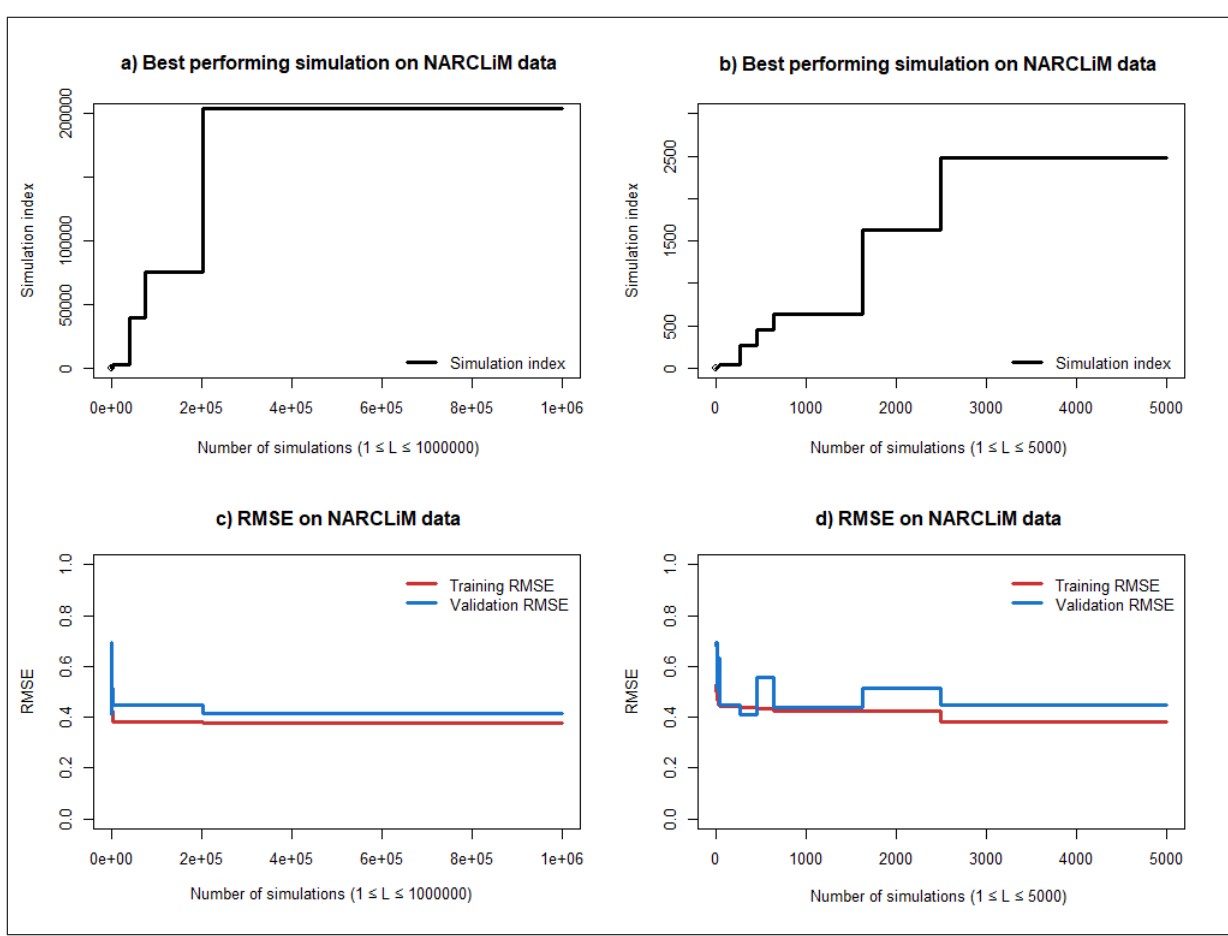

**Figure 1.** Sensitivity of the ensemble properties to the value of $L$. Left panes a) and c) contain results from all the simulations. Right panes b) and d) contain the results from the first 5000 simulations.

Though better RMSE results can be achieved with larger $L$, the marginal improvement in RMSE has high computational time cost. For the demonstration purposes in this study we select $L = 3000$ to accommodate for possible differences in RMSE changes between different datasets. As we will show below even with a sub-optimal value of $L$, MCE method has high performance and stable results.

### 2.2.2 Model interdependence

While we do not claim that the proposed method explicitly addresses the issue of model dependence, it is implicitly addressed to some degree at Step 3 in Table 2 of the MCE method. If there are two or more highly correlated models only one of them can be chosen at each step, and thus the resulting sum of such models' weights will be close to the scenario when only one of those models is kept in the ensemble.

We demonstrate this property of the MCE method on modified NARCliM data by adding a copy of one of the models with an added small random error and comparing the resulting weights as shown in Figure 2. To mitigate difference in weight values between random simulations we repeat the calculation 100 times and compare the mean values of the weights.

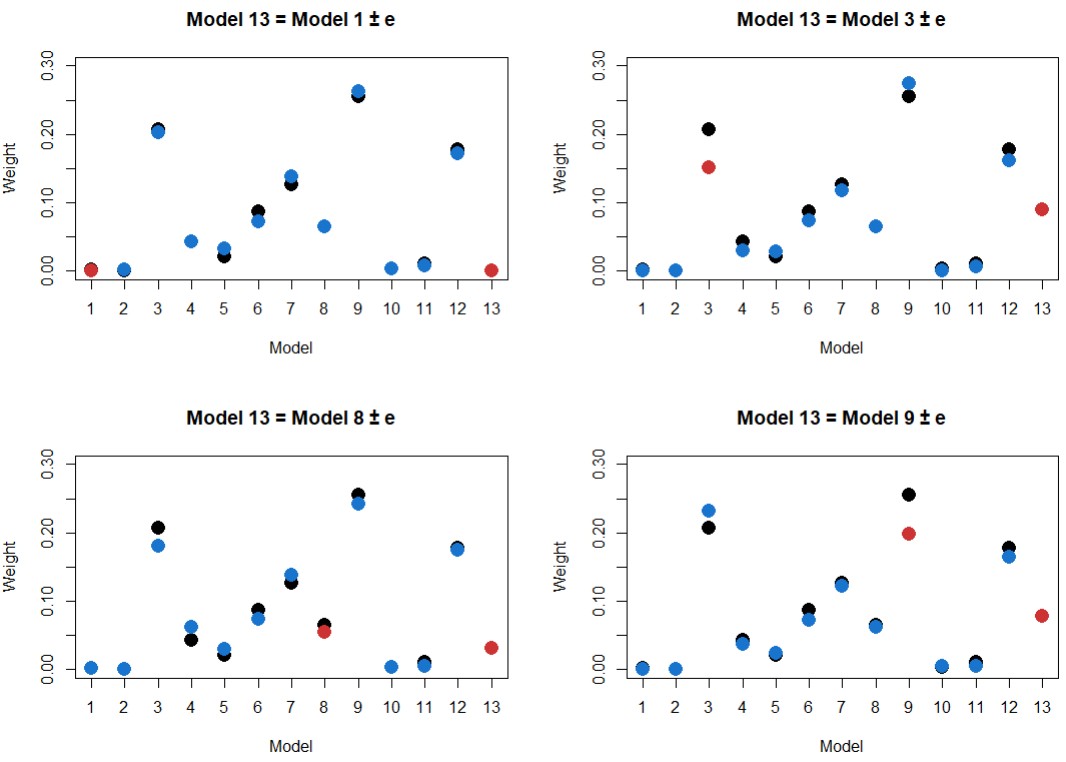

**Figure 2.** Change of MCE weights after adding a copy of Model 1, Model 3, 8 and 9 (clockwise from top left) to the NARCliM ensemble. The original MCE weights are in black. The weights of the modified ensemble are in blue, and the weights of the highly correlated models are in red.

As we can see from Figure 2, adding a highly correlated ensemble member does not significantly change the weights distribution significantly, and more pleasingly when a high performing model is duplicated, the weights are shared between the two copies (see Model 3 and Model 9). Consequently the performance of $E_{MCE}$ remains approximately the same. Though

we can not guarantee this behaviour in all types of data, we believe that the MCE method's design helps to mitigate the model interdependence problem.

### 2.2.3 MCE method limitations

Though the MCE method can be used on any climate dataset which contain the required inputs, its relative performance differs depending on the properties of the dataset. We will demonstrate that in the case of a normally distributed data, its performance is competitive with the simple averaging and other more sophisticated methods. In more challenging scenarios, when data is not normally distributed, MCE is performing better than the common alternatives.

As the MCE method is based on a stochastic process, the results between runs can vary. To mitigate this effect and to have
170 reproducible results we set the seed of R software's random number generator to a constant for all simulations. The MCE method in its current implementation does not provide an uncertainty quantification, and this limitation is a subject for future nonlinear ensemble weighting methods development.

Finally, as the MCE method does not consider spatial information, the resulting weights have limited physical interpretability. Extending the MCE method to utilize such information is a subject for future research.

### 175 2.3 Multi-model ensemble average (AVE) method

In order to evaluate the relative performance of the MCE method we select two other popular approaches to constructing ensemble weighted average. The first approach is the widely used average of individual climate model outputs (Lambert and Boer (2001); Gleckler et al. (2008)):

$$E_{AVE_t} = 1/N \sum_{j=1}^{N} M_{j,t}, \tag{3}$$

for each $1 \le t \le T$. If model differences from observations are random and independent, they will cancel on averaging and the resulting ensemble average will perform better than individual climate models (Lambert and Boer (2001)).

### 2.4 Convex optimisation (COE) method

The second approach that has been selected for relative performance evaluation in this study is a convex optimisation as proposed by Bishop and Abramowitz (2013). It represents a family of other methods based on a linear optimisation over the
185 vector space of individual climate model outputs.

The purpose of this method is to find a linear combination of climate model outputs with $w_1, w_2, ..., w_N$ weights which would minimise mean squared differences with respect to observations:

$$E_{COE_t} = \sum_{j=1}^{N} w_j M_{j,t}, \tag{4}$$

for each $1 \le t \le T$, so that $\sum_{t=1}^{T}(E_{COE_t} - O_t)^2$ is minimised under restrictions $\sum_{j=1}^{N} w_j = 1$ and $w_j \ge 0$ for each $1 \le j \le N$.

This method and its implementation are discussed in details in Bishop and Abramowitz (2013), and we show that it has relatively high performance on the chosen datasets. However, like any other linear optimisation technique, it naturally has some limitations that nonlinear optimisations like the MCE method do not. In particular, the COE method assumes having a large enough sample size to rule out spurious fluctuations in the weights associated with too small sample size. Such an assumption is not required for the MCE method. In addition, convex optimisation tends to set a large portion of weights equal to 0, as is shown in the examples below, which results in lower effective number of models used for prediction.

## 2.5 Performance metrics

### 2.5.1 RMSE

The root mean squared error (RMSE), Equation 5 is a frequently used measure of the differences between values (sample or population values) predicted by a model or an estimator and the values observed. RMSE is positive, and a value of 0 indicates a perfect fit to the data. In general, a lower RMSE is better than a higher one. However, comparisons across different types of data would be invalid because the measure is dependent on the scale of the numbers used. Minimising RMSE is commonly used for finding optimal ensemble weight vectors (e.g. Herger et al. (2018); Krishnamurti et al. (2000)).

$$RMSE = \sqrt{\frac{1}{T}\sum_{t=1}^{T}\left(\sum_{j=1}^{N} w_j M_{j,t} - O_t\right)^2}, \tag{5}$$

with $\sum_{j=1}^{N} w_j = 1$ and $w_j > 0$ for $j = 1, \ldots, N$. $T$ is the total number of time steps, $M_{j,t}$ denotes the value of model $j$ at time step $t$ and $O_t$ is the observed value at point $t$.

### 2.5.2 Trend bias

The monthly trend bias is calculated as the difference between the inclination parameter $a$ in weighted ensembles and observations estimated using a linear function $y = ax + b$ on validation data for each month. The total weighted ensemble trend bias metric is calculated as a mean of the monthly trend biases.

### 2.5.3 Climatology monthly bias

The monthly bias is calculated as the difference between the mean of the weighted ensemble and the observation for each month on validation data. The total climatology monthly bias metric is calculated as a mean of the monthly biases.

### 2.5.4 Interannual variability

Interannual variability for each month is calculated as the difference between the standard deviation of detrended weighted ensemble and the standard deviations of detrended observations on validation data. The total interannual variability metric is calculated as the mean of interannual variability for each month.

### 2.5.5 Climatological monthly RMSE

Climatological RMSE is calculated according to Equation 5 on climatological monthly means of weighted ensemble values and observations on validation data.

## 2.6 Cross-validation procedures

### 2.6.1 Holdout method

In this method the dataset, which contains the observations, is split into a training (or calibration) set and a validation (or testing) set. The goal of cross-validation is to examine the model's ability to predict new data that was not used in estimating the required parameters.

We partition our data into two sets, with 70% of data used for training and 30% for validation. This is a specific case of the K-fold validation procedure (Refaeilzadeh et al., 2009, p. 532-538), which is relatively simple to apply and discuss, facilitating the sharing of our findings with other members of the research and non-research communities.

### 2.6.2 Model-as-truth performance assessment

To evaluate each method's performance on the future model projections, we use the model-as-truth approach and analyse the metrics described in Section 2.5. At each step of model-as-truth performance assessment one model is selected as a true model (pseudo-observations) and the remaining models are used to build a weighted ensemble mean that best estimates the true model over the historical period. This weighted ensemble mean is then tested against the future projections of the true model. For a given ensemble this is repeated as many times as the number of the ensemble members with a different member being chosen as the true model each time. The median and spread of these results is reported.

## 3 Results

### 3.1 CMIP5 data

Though the selected monthly CMIP5 data contains annual variation, it is not predominant due to the length and trend of the dataset as shown in panel a) in Figure 3. The CMIP5 models output distribution is close to normal as shown in panel b) in Figure 3.

Applying the MCE method on the selected CMIP5 data with $T = 120$ (1900 - 2019) and a training period $T_1 = 80$ (1900 - 1979), we obtain a weighted ensemble mean $E_{MCE}$ and compare it with outputs from other methods. We summarize CMIP5 data properties together with the resulting ensemble's weights in Figure 3 and holdout cross-validation results in Table 3.

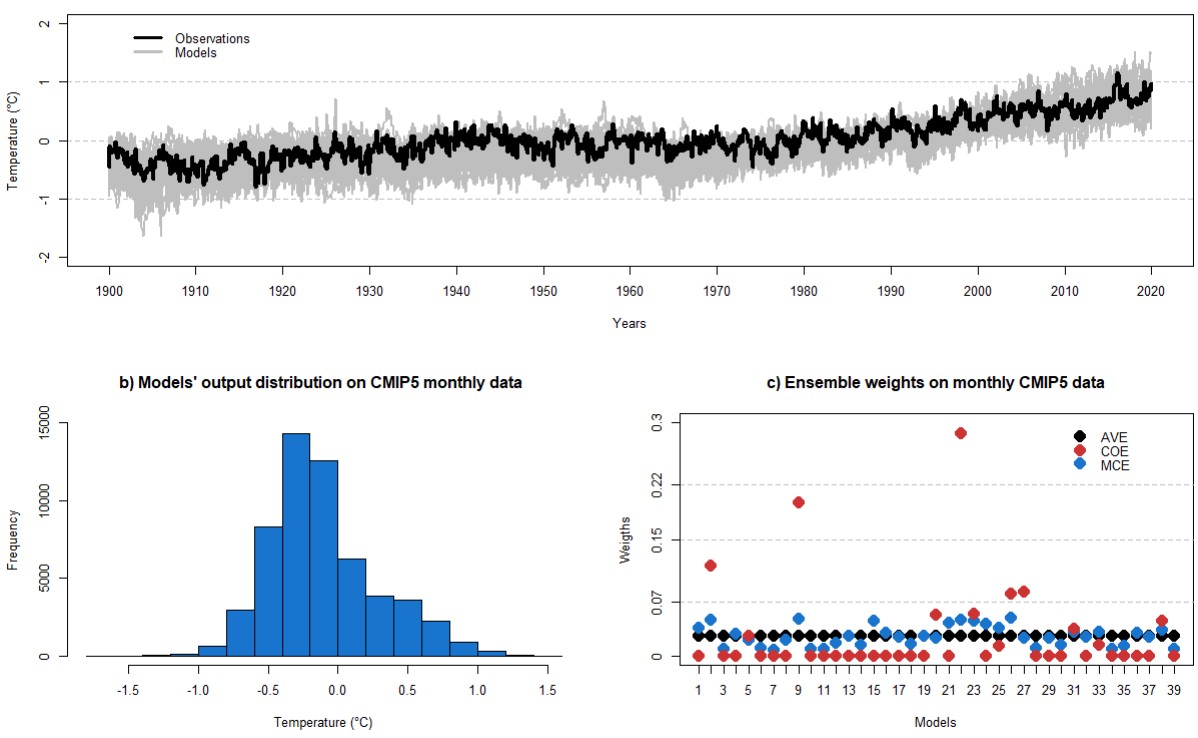

**Figure 3.** CMIP5 data properties. **a)** Model outputs and observations. **b)** Model output distribution. **c)** AVE, COE and MCE weights.

| $Ensemble$ | $RMSE_T$ | $RMSE_V$ | $B_T$ | $B_{CM}$ | $B_{IV}$ | $RMSE_{CM}$ |
|:---:|:---:|:---:|:---:|:---:|:---:|:---:|
| $E_{AVE}$ | 0.22 | 0.17 | 0.01 | -0.09 | -0.05 | 0.10 |
| $E_{COE}$ | 0.15 | 0.19 | 0.00 | -0.12 | -0.04 | 0.13 |
| $E_{MCE}$ | 0.18 | 0.17 | 0.01 | -0.10 | -0.05 | 0.10 |

**Table 3.** Performance comparison of different methods on CMIP5 data, RMSE on training ($RMSE_T$) and validation ($RMSE_V$) data; trend bias ($B_T$), climatological monthly bias ($B_{CM}$), interannual variability bias ($B_{IV}$) and climatological monthly RMSE ($RMSE_{CM}$) on validation data.

We can see that $E_{AVE}$ and $E_{MCE}$ perform at a similar RMSE level, with $E_{COE}$ performance decreasing comparatively more in validation, a possible indication of overfitting to the training data. We can see from Figure 3 that the COE method tends to set zero weights to some models, but builds a weighted ensemble mean that performs best on the training period (1900-1979). Due to some models having zero weights, some of the models' diversity is lost, and this results in worse performance on the validation period ($RMSE_V$ and $RMSE_{CM}$ in Table 3). The MCE method, on the other hand, produces model weights that

vary around $1/N$, where $N$ is the number of the models. The MCE method does not give any model zero weighting and hence preserves the ensembles' diversity. The climatological biases $B_T$, $B_{CM}$ and $B_{IV}$ are nearly equal for all three methods.

The model-as-truth performance assessment is done on $T = 200(1900-2099)$ and a training period $T_1 = 120(1900-2019)$ as described in Section 2.6.2. The results are summarized in Figure 4 and Table 4 in form of median, 25% and 75% percentiles of the $N = 39$ (number of models) values.

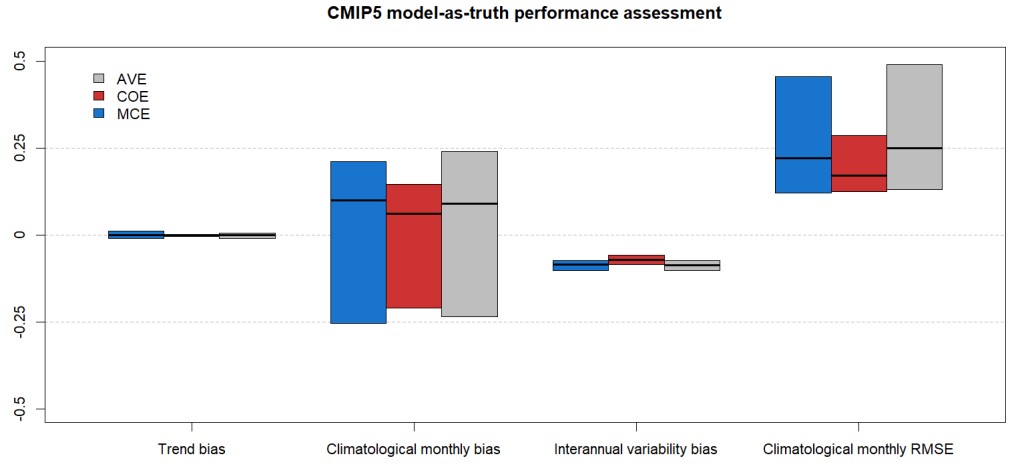

**Figure 4.** CMIP5 model-as-truth performance assessment results. Median, 25% and 75% percentiles of $N = 39$ models.

| Ensemble | $B_T$ | $B_{CM}$ | $B_{IV}$ | $RMSE_{CM}$ |
|:---:|:---:|:---:|:---:|:---:|
| $E_{AVE}$ | 0.00 | 0.09 | -0.09 | 0.25 |
| $E_{COE}$ | 0.00 | 0.06 | -0.07 | 0.17 |
| $E_{MCE}$ | 0.00 | 0.10 | -0.09 | 0.22 |

**Table 4.** Model-as-truth performance comparison of different methods on CMIP5 data, median of trend bias ($B_T$), climatological monthly bias ($B_{CM}$), interannual variability bias ($B_{IV}$) and climatological monthly RMSE ($RMSE_{CM}$) on validation data.

All the methods perform similarly in model-as-truth assessment with $E_{COE}$ having better $RMSE_{CM}$.

## 3.2    NARCliM data

The seasonal variation in NARCLiM data is larger than in CMIP5 data as shown in panel a) in Figure 5. The NARCLiM models output distribution is not normal as shown in panel b) in Figure 5 due to summer time and winter time temperature peaks.

We apply the MCE method on the selected NARCLiM data with $T = 20$ (1990 - 2009) and a training period $T_1 = 14$ (1990 - 2003), obtain a weighted ensemble mean $E_{MCE}$ and compare it with outputs from other methods. We summarize NARCLiM data properties together with the resulting ensemble's weights in Figure 5 and holdout cross-validation results in Table 5.

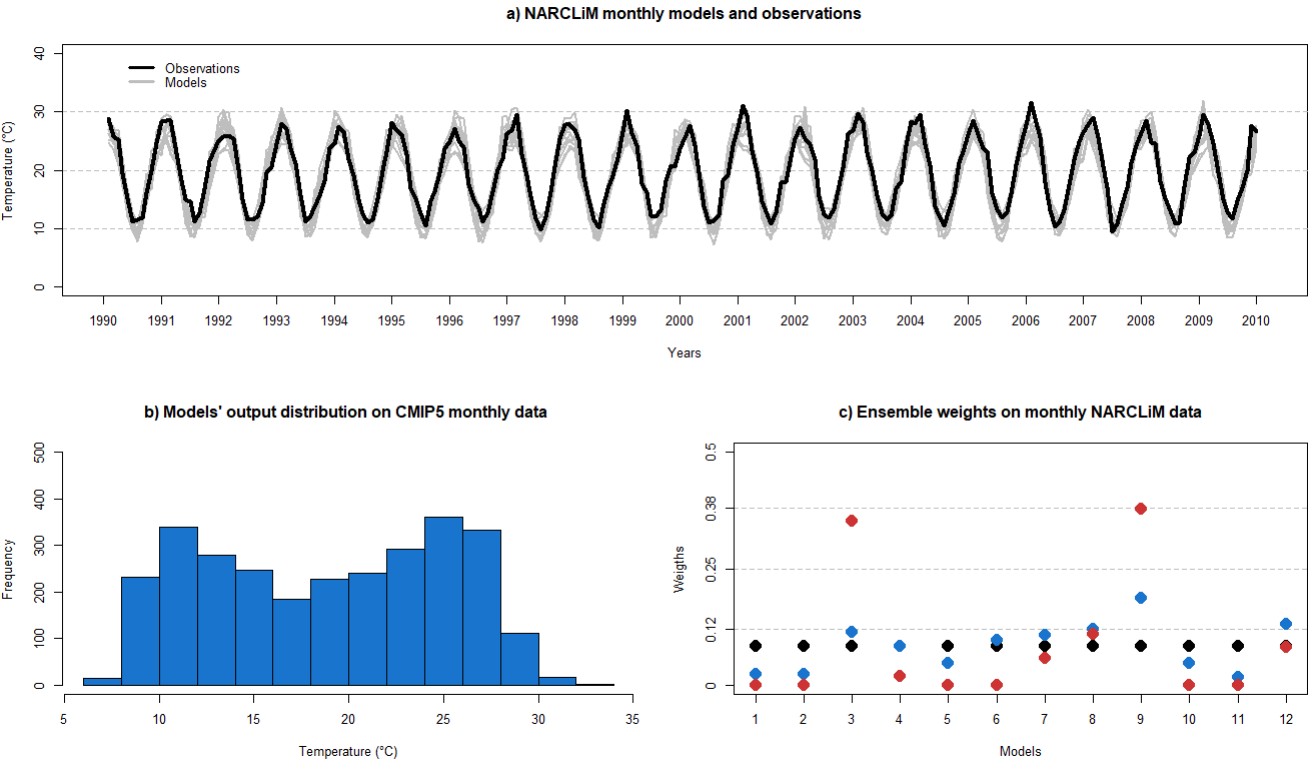

**Figure 5.** NARCLiM data properties. **a)** Model outputs and observations. **b)** Model output distribution. **c)** AVE, COE and MCE weights.

| $Ensemble$ | $RMSE_T$ | $RMSE_V$ | $B_T$ | $B_{CM}$ | $B_{IV}$ | $RMSE_{CM}$ |
|:---:|:---:|:---:|:---:|:---:|:---:|:---:|
| $E_{AVE}$ | 1.6 | 1.85 | 0.04 | -1.16 | -0.73 | 1.19 |
| $E_{COE}$ | 1.32 | 1.58 | 0.00 | -0.49 | -0.59 | 0.64 |
| $E_{MCE}$ | 1.4 | 1.58 | 0.04 | -0.64 | -0.69 | 0.70 |

**Table 5.** Performance comparison of different methods on NARCLiM data, RMSE on training ($RMSE_T$) and validation ($RMSE_V$) data; trend bias ($B_T$), climatological monthly bias ($B_{CM}$), interannual variability bias ($B_{IV}$) and climatological monthly RMSE ($RMSE_{CM}$) on validation data.

As in CMIP5 data analysis (Figure 3), we see that the MCE method is maintaining (i.e., assigning non-zero weights to) more models in the final weighted ensemble than the COE method. As the number of models is significantly smaller than in CMIP5

case, the difference between the MCE output weights and the equal weights is also considerably larger. The MCE method shows itself capable of maintaining much of the ensembles' diversity during the optimization process. This allows MCE to substantially improve performance over the AVE method on both training and validation periods and perform at the same level as COE on validation period even with lower $RMSE_T$. Again, COE has a larger decline in performance from training to validation periods indicating possible overfitting.

The model-as-truth performance assessment is done on $T = 30$ (1990 - 2019 and 2030-2039) and a training period $T_1 = 20$ (1990 - 2019) as described in Section 2.6.2. The results are summarized in Figure 6 and Table 6 in form of median, 25% and 75% percentiles of the $N = 12$ (number of models) values.

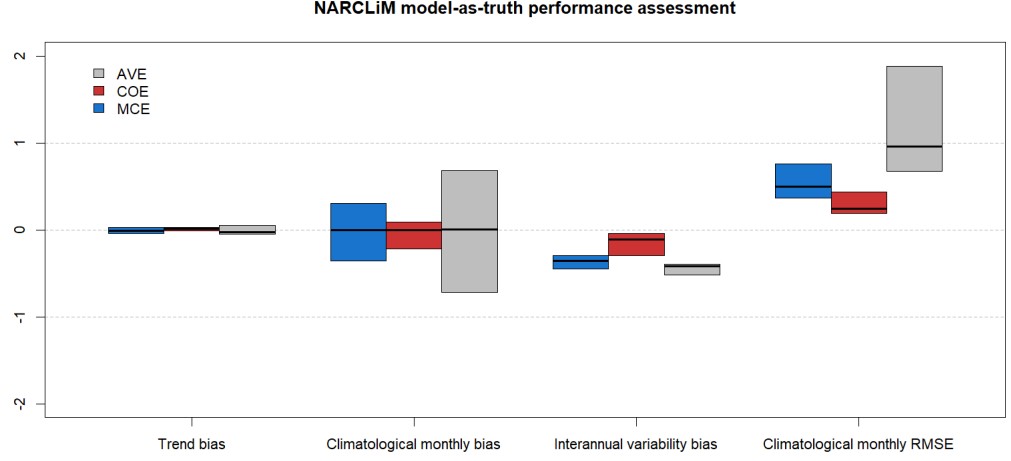

**Figure 6.** NARCLiM model-as-truth performance assessment results. Median, 25% and 75% percentiles of the $N = 12$ models.

| Ensemble | $B_T$ | $B_{CM}$ | $B_{IV}$ | $RMSE_{CM}$ |
|----------|-------|----------|----------|-------------|
| $E_{AVE}$ | -0.03 | 0.01 | -0.42 | 0.96 |
| $E_{COE}$ | 0.01 | -0.01 | -0.11 | 0.24 |
| $E_{MCE}$ | -0.02 | -0.01 | -0.36 | 0.50 |

**Table 6.** Model-as-truth performance comparison of different methods on NARCLiM data, median of trend bias ($B_T$), climatological monthly bias ($B_{CM}$), interannual variability bias ($B_{IV}$) and climatological monthly RMSE ($RMSE_{CM}$) on validation data.

As in the CMIP5 results (Figure 4 and Table 4) all the methods perform at the same level in $B_T$ and $B_{CM}$ metrics of the model-as-truth assessment. In $B_{IV}$ and $RMSE_{CM}$ metrics $E_{COE}$ performs better, while $E_{AVE}$ performs worse than $E_{MCE}$.

### 3.3 KMA data

The KMA data is non-negative with a non-normal distribution of model outputs and observations as shown in panels a) and b) in Figure 7.

Applying the MCE method on the selected data with $T = 33$ (1973 - 2005) and a training period $T_1 = 22$ (1973 - 1994), we obtain a weighted ensemble mean $E_{MCE}$ and compare it with outputs from other methods. As KMA data contains only summertime months, we analyse only its $RMSE_T$ and $RMSE_V$. We summarize KMA data properties together with the resulting ensemble's weights in Figure 7 and holdout cross-validation results in Table 7.

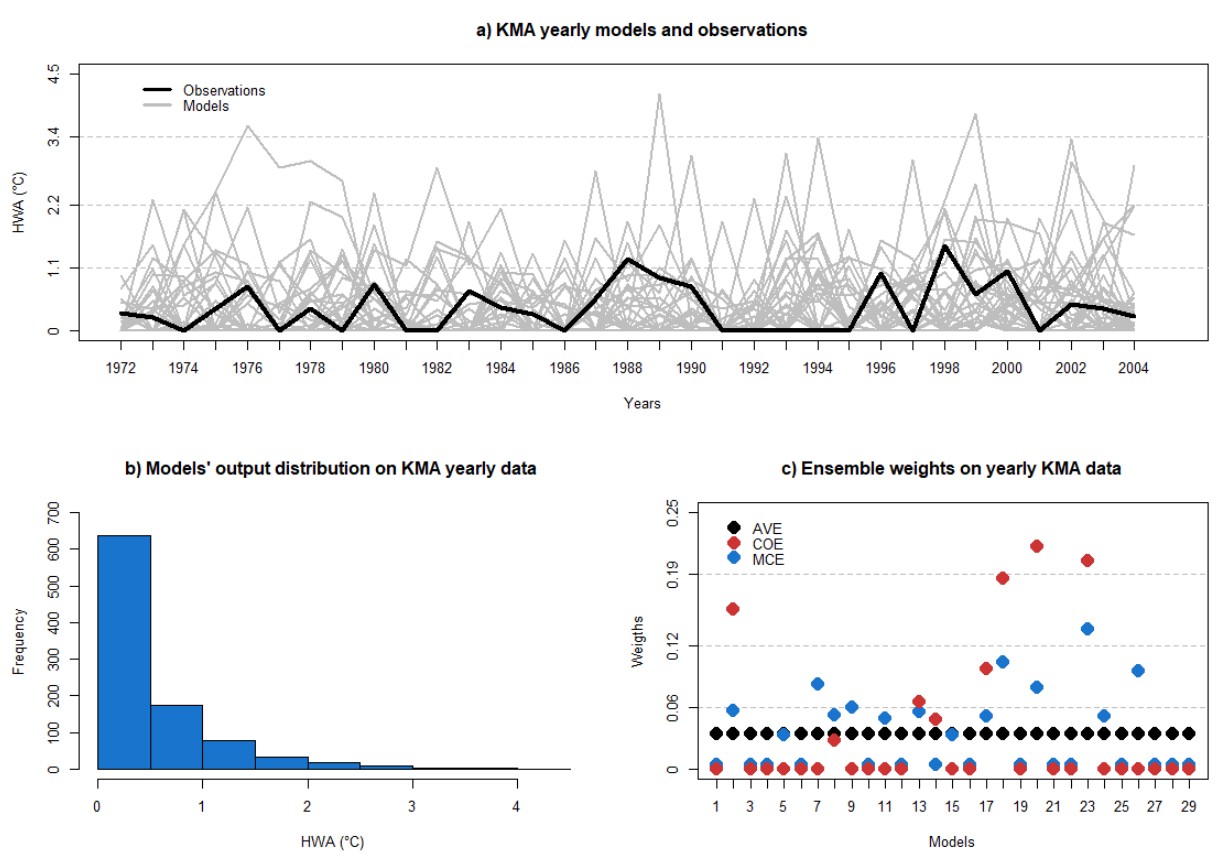

**Figure 7.** KMA data properties. **a)** Model outputs and observations. **b)** Model output distribution. **c)** AVE, COE and MCE weights.

| $Ensemble$ | $RMSE_T$ | $RMSE_V$ |
|:---:|:---:|:---:|
| $E_{AVE}$ | 0.36 | 0.5 |
| $E_{COE}$ | 0.23 | 0.52 |
| $E_{MCE}$ | 0.29 | 0.44 |

**Table 7.** Performance comparison of different methods on KMA data, RMSE on training ($RMSE_T$) and validation ($RMSE_V$) data.

We can see that MCE has the lowest RMSE and maintains the ensembles' diversity with a few models receiving zero weights. The COE method gives non-zero weights to only a small subset of models, which results in its performance on the validation period being lower compared to MCE.

## 4 Discussion

The obtained results indicate that Markov chains can be used to construct a better performing weighted ensemble mean with lower RMSE on validation data than commonly used methods like multi-model ensemble average and convex optimisation (Tables 3, 5 and 7). As the method's performance did not degrade from training to validation as much as COE, we are confident that it is less prone to over-fitting than linear optimisation methods. We attribute this advantage of the MCE method to its ability to maintain the ensemble's diversity while optimising its weights on the training period (Figures 3, 5 and 7), to mitigate model interdependence and to capture some of the nonlinear patterns in the data.

The MCE method also performs at the same level as other methods in terms of climatological metrics and model-as-truth performance assessment, which gives us confidence in its ability to be used for future estimation of climate variables.

However, as previous studies show (e.g. Masson and Knutti (2011); Sanderson et al. (2017)) and as discussed in Section 2.2.3, extending the MCE method to include spatial information would improve our ability to interpret the physical meaning of the resulting weights.

As the number of models increases, MCE tends to become closer to AVE weights (Figure 3), while being closer to COE with a smaller number of models (Figure 7). This phenomenon can be explained by a higher effect of diversity on performance in larger ensembles with normally distributed data (observations and model outputs) than in smaller ensembles like NAR-CLiM. The KMA data has an intermediate number of models and MCE produces a hybrid response which maintains ensemble diversity (a few models with zero weights) but does weight a small number of models more highly.

The MCE method is computationally cheap and is limited only by a software's ability to handle extreme numerical values. One limitation of the MCE method is its current inability to quantify the uncertainty of the resulting weighted ensemble mean. However, we believe that given the stochastic nature of the method, this limitation can be overcome in future implementations. MCE performance can be further improved by combining it with other types of optimisation, e.g. linear. In addition, other nonlinear optimisation techniques, which would include more complex structures than simple Markov chains, can be developed based on our demonstrated results.

Finally, the MCE method doesn't require some of the assumptions necessary for the multi-model ensemble average method (e.g. models being reasonably independent and equally plausible as discussed by Knutti et al. (2017)) and it doesn't produce as many zero weights as the convex optimisation method, hence maintaining more of the models' diversity. We attribute the tendency of the COE method to set zero weights to some models to its property below:

Geometrically, the restrictions $w_j \geq 0, \sum_{j=1}^{N} w_j = 1$ describe a simplex in $R^N$ that is a subset of the hyperplane with the equation $\sum_{j=1}^{N} w_j = 1$. Denote $w = (w_1, w_2, \ldots, w_N)$. The potential choice of weights that only satisfy the constraint $\sum_{j=1}^{N} w_j = 1$ without the non-negativity restriction represents any point in the hyperplane $P = \{w : \sum_{j=1}^{N} w_j = 1\}$. This hyperplane contains the simplex $S = \{w \in P : w_j \geq 0\}$. In general, the optimal point $w^*$ for the unrestricted solution of the optimisation problem

$$\min_{w} \sum_{i=1}^{T} (\sum_{j=1}^{N} w_j M_{j,i} - O_i)^2, w \in P$$

will be outside the simplex. It is clear that the optimal point for the constrained solution on the simplex:

$$\min_{w} \sum_{i=1}^{T} (\sum_{j=1}^{N} w_j M_{j,i} - O_i)^2, w \in S$$

would be on the boundary of the simplex rather than in its interior. Indeed, if we assume that the optimal point for the constrained problem is certain $\tilde{w}$ in the interior of the simplex, we immediately arrive at a contradiction. Take then the point $\hat{w} = w^* + \lambda(\tilde{w} - w^*)$ with $\lambda \in (0,1)$ chosen such that $\hat{w}$ is on the intersection of the line connecting $w^*$ and $\tilde{w}$ with the boundary of the simplex. Because of the strict convexity of the function

$$f(w) = \sum_{i=1}^{T} (\sum_{j=1}^{N} w_j M_{j,i} - O_i)^2$$

we have:

$$f(\hat{w}) = f(w^* + \lambda(\tilde{w} - w^*)) = f(\lambda\tilde{w} + (1-\lambda)w^*) < \lambda f(\tilde{w}) + (1-\lambda)f(w^*) < f(\tilde{w})$$

in contradiction to the assumption that $\tilde{w}$ delivers the minimum over the simplex. Hence the optimisation on the simplex tends to deliver optimal points with some components equal to zero because they tend to be on the boundary of the simplex.

## 5 Conclusions

In this study, we presented a novel approach based on Markov chains to estimate model weights in constructing weighted climate model ensemble means. The complete MCE method was applied to selected climate datasets, and its performance was compared to two other common approaches (AVE and COE) using cross-validation holdout method and model-as-truth performance assessment with RMSE, trend bias, climatology monthly bias, interannual variability and climatological monthly RMSE metrics. The MCE method was discussed in detail, and its step-wise implementation, including mathematical background, was presented (Table 2).

The results of this study indicate that applying nonlinear ensemble weighting methods on climate datasets can improve future climate projection in terms of accuracy. Even a simple nonlinear structure such as Markov chains shows good performance on different commonly-used datasets compared to linear optimisation approaches. These results are supported by using standard performance metrics, cross-validation procedures and model-as-truth performance assessment. The developed MCE method is objective in terms of parameter selection, has a sound theoretical basis and has a relatively low number of limitations. It maintains ensemble diversity, mitigates model interdependence and captures some of the nonlinear patterns in the data while optimizing ensemble weights. It is also shown to perform well on non-Gaussian datasets. Based on the above, we are confident to suggest its application on other datasets and its usage for the future development of new nonlinear optimisation methods for weighting climate model ensembles.

*Code and data availability.* Code and data for this study is available at https://doi.org/10.5281/zenodo.4548417.

*Author contributions.* All co-authors contributed to method development, theoretical framework and designing of experiments. MK, YF and JPE selected climate data for this study. MK developed the model code with contribution from SP and performed the simulations. RO prepared NARCLiM and KMA data. MK and YF prepared the manuscript with contribution from all co-authors.

*Competing interests.* The authors declare no competing interests.

*Acknowledgements.* MK would like to acknowledge the support from the UNSW Scientia PhD Scholarship Scheme. RO would like to acknowledge the support from the National Research Foundation of Korea (NRF) grant funded by the Korea government (MSIT) (NRF-2018R1A5A1024958).

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
