# Peer review of "A Markov chain method for weighting climate model ensembles"

_Geoscientific Model Development, 2020_

## Referee Comment (RC1) · Ben Sanderson (Referee) · 2 Nov 2020

This study considers a novel application of Markov Chain methods to the problem of climate model weighting. The authors use a multi-model ensemble of climate model simulations, together with a range of different observation sources (global mean temperatures, regional averages and regional extreme temperature statistics). The novel "MCE" methodology is compared with two other approaches (a climatological weighted average score - Lambert and Boer 2001, AVE hereafter, and the ensemble transformation approach of Bishop and Abramowitz 2013, COE hereafter). The authors find desirable properties of their proposed approach in terms of out-of-sample skill and performance in non normally distributed quantities related to temperature extremes.

[Figure]

The approach shows promise, the potential for more robust error estimates in weights would be useful and the stochastic nature of the computation may offer additional benefits. However, these avenues are not explored in the current text and the authors have not yet fully addressed the basic requirements of an operational climate model weighting scheme (robust out of sample testing), and existing simpler approaches may outperform the approach presented here for the metrics considered (see below).

The introduction talks of the need to represent and consider model interdependency - but the actual study does not propose any mechanism for accounting for model interdependencies or common components in the weighting scheme. Example approaches for doing so are laid out in Sanderson (2017) and Lorenz (2018).

The study also generalizes existing literature by the two comparative cases considered (AVE and COE) as "linear" approaches, but there are more complex schemes in the literature which address issues not considered here, like optimal subselection (Sanderson 2015). In addition, given an RMSE or $R^2$ metric - methods which determine the global optimization of weighted scores (Herger 2018) are in theory unbeatable - so, although there may be tangential benefits to using MCE, it's unclear that the approach here could outperform a global weight optimization for these types of metric.

The paper attributes differences in weighting behavior between MCE, AVE and COE to differences in methodology, without sampling the subjective degrees of freedom in each of the approaches. As such, the observation that the MCE approach tends to rule out fewer models than COE is potentially a function of the chosen L and sigma parameters as well as the MCE approach itself. Any revision should present parameter sensitivities in the main text.

The paper also employs only a weak out of sample test, splitting the available data into a training and validation set. This is insufficient for the climate problem - where only the past is observable and the primary unknowns are climate projections in the future. Models with comparatively similar past trajectories might diverge significantly in

the future - and the validation scheme considered here (where random timesteps are withheld) does not capture this divergence. A stronger test is a perfect model study, where individual models are withheld from the ensemble and late century projected performance is considered.

The paper considers that, for a given output variable, that the climate models should be weighted only by their fidelity in producing that variable - but there is no clear reason why this should be the case. Climate projections are functions of the integrated climate system which determine global climate sensitivities and regional feedbacks. As such, fidelity in producing a given variable (such as a regional temperature timeseries) in the past is no guarantee of accuracy in the future (Sanderson 2012). Lorenz et al (2018), for example, discusses at length the relative utility of different variable types for constraining future model evolution.

Finally, the paper does not acknowledge the various reasons climate models may disagree - natural variability provides an absolute limit on the performance of an uninitialized climate model, and so some discussion is required on how different sources of error would impact the results in this case.

Given these issues, I suggest the following revisions:

1 - outline more clearly (and ideally quantify) the benefits of using MCE beyond absolute skill scores (where the method will be outperformed by construction by Herger 2018)

2 - present the parameter sensitivity of the method in the main text

3 - perform a rigorous leave-one-out test of weighting scheme performance in a perfect model projections of century-scale climate change

4 - Consider how to address model interdependency in this method (which would bias the leave-one-out test)- either by adaptation of existing approaches or otherwise

5 - Reconsider the relationship between variables used for weighting and those variables which are to be weighted (there is no reason to limit consideration in the weighting term to those variables which are themselves being weighted).

6 - consider the role of various sources of error and whether they should be represented within the scheme (natural variability, forcing errors, structural differences).

References:

Lorenz, R., Herger, N., Sedláček, J., Eyring, V., Fischer, E. M., & Knutti, R. (2018). Prospects and Caveats of Weighting Climate Models for Summer Maximum Temperature Projections Over North America. J. Geophys. Res. Atmos., 123(9), 4509–4526. doi: 10.1029/2017JD027992

Herger, N., Abramowitz, G., Knutti, R., Angélil, O., Lehmann, K., & Sanderson, B. M. (2018). Selecting a climate model subset to optimise key ensemble properties. Earth Syst. Dyn., 9(1), 135–151. doi: 10.5194/esd-9-135-2018

Bishop, C. H., & Abramowitz, G. (2013). Climate model dependence and the replicate Earth paradigm. Clim. Dyn., 41(3), 885–900. doi: 10.1007/s00382-012-1610-y

Lambert, S. J., & Boer, G. J. (2001). CMIP1 evaluation and intercomparison of coupled climate models. Clim. Dyn., 17(2), 83–106. doi: 10.1007/PL00013736

Sanderson, B. M., & Knutti, R. (2012). On the interpretation of constrained climate model ensembles. Geophys. Res. Lett., 39(16). doi: 10.1029/2012GL052665

Sanderson, B. M., Wehner, M., & Knutti, R. (2017). Skill and independence weighting for multi-model assessments. Geosci. Model Dev., 10(6), 2379–2395. doi: 10.5194/gmd-10-2379-2017

Sanderson, B. M., Knutti, R., & Caldwell, P. (2015). A Representative Democracy to Reduce Interdependency in a Multimodel Ensemble. J. Clim., 28(13), 5171–5194. doi: 10.1175/JCLI-D-14-00362.1

---

## Referee Comment (RC2) · Anonymous Referee #2 · 10 Nov 2020

In this study, the authors present a novel method that employs Markov Chains as a means to weight members of global climate model (GCM) ensembles. Using three case studies involving historical simulations of global average temperature, regionally downscaled seasonal temperature, and a regional heat wave heuristic, they compare the performance of three model weighting schemes: simple model averaging (by definition, equal weights), a 'Convex Optimization Ensemble' (COE) method, and their 'Markov Chain Ensemble' (MCE) approach. Standard observational datasets from the recent past (up to ∼120 years) are used for comparison based on RMSE and R2 skill scores.

The proposed approach is interesting, and could be quite useful as a means to weight model ensembles and its simplicity is attractive, while also presenting a less ad-hoc approach than simple model averaging. However, I cannot evaluate the scientific merit of the approach because the model-observation tests are ill-posed in their current form. The main problem is that comparing unfiltered GCM time series to observations is very problematic when applying typical skill scores because the interannual variability will not correspond between CMIP5 models and the observations. So although the underlying trend or evolving signal (assuming there is a signal, such as in global temperatures) of a perfectly performing model should match what is observed, the full time series from the model would not necessarily match the observed time series. This is because in the CMIP5 experiments, the GCMs begin the experiments with different initial conditions, and different model runs within the same model will begin at different points in the same control run before then beginning the perturbation experiment (i.e. including anthropogenic forcings). This makes direct time series comparisons very tricky if not handled carefully. Take a very simple example as shown in the example figure.

Here are three simulated white noise time series (mean = 0, standard deviation = 1) overlaid onto two linear trends (0.1 for the black and blue time series, 0.03 for the red time series). The blue and red time series symbolize what could occur in a multi-member CMIP5 ensemble. Note that even though the blue time series has exactly the same trend as the 'observed' time series, the RMSE is higher than the red time series simply because the inter-annual variation is misaligned (of opposite sign in this case) with the observed anomalies. In contrast, the red time series has a lower RMSE, despite the fact that it does not capture the true forced trend. But the anomalies are aligned perfectly with observations. The red time series would be weighted higher in this case. It is 'right' for the 'wrong' reasons. Similarly the test set up in this manuscript is subject to the same problem. Individual models could exhibit anomalies that are more similar to the observed time series (driving down the skill scores), while the model response to the perturbation is less accurately simulated than other models with higher (by-chance) anomaly errors.

Of course, all the methods that were tested (AVG, COE, MCE) could be similarly bi-
ased, in which case perhaps the results hold. But, there is no way to ascertain that in
the current test structure.

My other, more minor comments relate to similar issues with the structure of the model
evaluation exercise. It is unsurprising that the model RMSE and R2 values were so
poor when comparing GCM results to heat wave heuristics based on local weather sta-
tion data (and again, where the model internal variability would have no reason except
by chance to match the observed internal variability). The GCMs were developed at
a scale that was never intended to resolve such localized patterns, and of course any
annual heat waves that were observed, would only by chance occur in the same years
(and be of similar magnitude) in the ensemble members.

To improve the evaluation I suggest the authors revisit the literature to see how oth-
ers have tackled this problem. More attention should be paid to, for example, efforts
by Sanderson et al. (2015) to carefully construct valid comparisons between GCM
ensembles and observations (in this case, by focusing the model skill evaluation on
climatologies, rather than time series anomalies) while also taking into account model
inter-dependence; something which the authors admit they do not account for in their
method. Others have addressed the non-initialized climate model/observation com-
parison problem by comparing long-term trends (e.g. Terando et al. 2012), which
removes some of the problems with mis-matched internal variability, and gets closer to
an actual forecast verification approach, but does not address other issues such as the
robustness and reliability of weighting methods (see discussion in Knutti et al. 2017). It
should be possible to construct a rigorous test of their MCE method, but the numerous
challenges that have been widely and repeatedly documented in the literature should
be acknowledged and addressed.

References:

Knutti, R., J. Sedláček, B. M. Sanderson, R. Lorenz, E. M. Fischer, and V. Eyring,

2017: A climate model projection weighting scheme accounting for performance and interdependence. Geophys. Res. Lett., 44, 1909–1918, doi:10.1002/2016GL072012. http://doi.wiley.com/10.1002/2016GL072012.

Sanderson, B. M., R. Knutti, and P. Caldwell, 2015: Addressing Interdependency in a Multimodel Ensemble by Interpolation of Model Properties. J. Clim., 28, 5150–5170, doi:10.1175/JCLI-D-14-00361.1. http://journals.ametsoc.org/doi/abs/10.1175/JCLI-D-14-00361.1.

Terando, A., K. Keller, and W. E. Easterling, 2012: Probabilistic projections of agro-climate indices in North America. J. Geophys. Res., 117, D08115, doi:10.1029/2012JD017436. http://doi.wiley.com/10.1029/2012JD017436.

[Figure]

**Fig. 1.** RMSE Example

[Figure]

---

## Author Comment (AC1) · 14 Jan 2021

We provide a point by point response to the referee's comments. Referee's original comments are in blue fonts, and our response in black font. The figures from the new version of the publication are included in the end of this document for reference.

Referee 1: Ben Sanderson sanderson@cerfacs.fr

This study considers a novel application of Markov Chain methods to the problem of climate model weighting. The authors use a multi-model ensemble of climate model simulations, together with a range of different observation sources (global mean temperatures, regional averages and regional extreme temperature statistics). The novel "MCE" methodology is compared with two other approaches (a climatological weighted average score - Lambert and Boer 2001, AVE hereafter, and the ensemble transformation approach of Bishop and Abramowitz 2013, COE hereafter). The authors find desirable properties of their proposed approach in terms of out-of-sample skill and performance in non normally distributed quantities related to temperature extremes.

The approach shows promise, the potential for more robust error estimates in weights would be useful and the stochastic nature of the computation may offer additional benefits. However, these avenues are not explored in the current text and the authors have not yet fully addressed the basic requirements of an operational climate model weighting scheme (robust out of sample testing), and existing simpler approaches may outperform the approach presented here for the metrics considered (see below).

Response:

Thank you for this suggestion, and we have now added additional model-as-truth (in- and out-of-sample) performance assessments in Section 4 ("Model-as-truth performance assessment"). The assessment is done on CMIP5 monthly data to increase the sample size and contains both performance (RMSE) and climatology (mean, trend) metrics.

We included the following text starting from line 243:

To evaluate the MCE method performance on the future model projections, we assess it using model-as-truth approach and compare its performance in terms of accuracy, trend and climatology to the multi-model ensemble average. We use one run per each CMIP5 model (39 in total) to reduce the number of highly correlated members in the ensemble. This approach is consistent with the one used in Herger et al. (2018); Sanderson et al. (2017) and other studies. At each step one model is selected as a true model and the remaining models are used to build $E_{AVE}$ and $E_{MCE}$. We use 1900-1979 period for training and 1980-2019 period for validation in historical climatology analysis and 1900 - 2019 period for training and 2020-2100 period for validation in future climatology analysis. To obtain a larger sample set we use monthly data and summarize the analysis results in Figure 7.

The temperature mean values are calculated over the whole period. The trends are calculated as linear approximation results at the end of the validation periods. The RMSE values are calculated are calculated on validation periods only. All the results are compared to the chosen model-as-truth ($O^*$), and the distances to $E_{AVE}$ and $E_{MCE}$ are used to calculate the improvement values: $(|E_{AVE} - O^*| - |E_{MCE} - O^*|)/|E_{AVE} - O^*|$. The results are summarized in terms of median, 25% and 75% percentiles.

The mean and trend improvement values on historical and future period indicate that the results discussed in Section 3.1 are robust in terms of climatology compared to the multi-model ensemble average. The $E_{MCE}$ RMSE improvement values for the historical period are on par with $E_{AVE}$, and we attribute this to the similar distribution of model weights on large model sets as shown in Figure 4 for the yearly averages. The future RMSE improvement values are slightly higher compared to the historical period, and we attribute it to the longer training data set, which allows the MCE method to better capture the non-linear dependencies between the ensemble members.

As discussed in Section 2.2.2 we expect the MCE method to perform better on data that has less data points and is not normally distributed. To evaluate this statement we perform model-as-truth performance assessment on monthly NARCLiM data in the same way as we did for CMIP5. We use all 12 models to maintain a big enough sample size. The historical climatology analysis include years 1990-2009 and years 1990-2003 for training. The future climatology analysis include years 1990-2009 used for training and 2020-2039 used for validation. The analysis is summarized on Figure 8.

The mean, trend and RMSE values on both periods indicate that the MCE method has higher performance on data that has a small number of data points, non normal distribution and high model interdependence (as shown on Figure 1). This is in line with the Section 3 results when comparing Table 2 and Table 3 values.

The introduction talks of the need to represent and consider model interdependency - but the actual study does not propose any mechanism for accounting for model interdependencies or common components in the weighting scheme. Example approaches for doing so are laid out in Sanderson (2017) and Lorenz (2018).

We added a new Section 2.7 ("Model interdependence") to discuss this point.

We included the following text starting from line 203:

While we do not claim that the proposed method explicitly addresses the issue of model dependence, it is implicitly addressed to some degree at Step 3 in Table 1 of the MCE method. If there are two or more highly correlated models only one of them will be chosen, and thus the resulting sum of such model's weight will be equal to when keeping only one of those models in the ensemble.

We demonstrate this property of the MCE method on modified KMA data by adding a copy of one of the models with an added small random error and comparing the resulting weights as shown in Figure 3.

Adding an ensemble member does not significantly change weights distribution, and consequently the performance of $E_{MCE}$ remains approximately the same. Though we can not guarantee this behaviour in all types of data, we believe that the MCE method's design helps to mitigate model's interdependence problem.

The study also generalizes existing literature by the two comparative cases considered (AVE and COE) as "linear" approaches, but there are more complex schemes in the literature which address issues not considered here, like optimal subselection (Sanderson 2015). In addition, given an RMSE or Rˆ2 metric - methods which determine the global optimization of weighted scores (Herger 2018) are in theory unbeatable - so, although there may be tangential benefits to using MCE, it's unclear that the approach here could outperform a global weight optimization for these types of metric.

The size limit of the publication does not allow to compare the MCE method to all the existing ensemble weighting techniques, though it would be a valuable addition to the study results. In addition, despite the fact that the MCE technique can be easily outperformed on training data by other methods (as shown in Tables 2,3 and 4), when it comes to cross-validation the MCE performs generally well on validation data (on par or better than other methods). We attribute its high performance to its ability to capture some of the sequential information in the input time series, which is not normally used by other methods. A simplified example of such sequential information would be: if model A is the closest to the observation at time t, then model B will be the closest at time t+1 with high probability. This would be naturally captured by the MCE method through the transition matrix P, but difficult (or impossible) to capture by linear or subselection methods.

The paper attributes differences in weighting behavior between MCE, AVE and COE to differences in methodology, without sampling the subjective degrees of freedom in each of the approaches. As such, the observation that the MCE approach tends to rule out fewer models than COE is potentially a function of the chosen L and sigma parameters as well as the MCE approach itself. Any revision should present parameter sensitivities in the main text.

We thank the referee for the above suggestion. We have now modified the MCE method to search for best parameters without a need for manual tuning. The modified algorithm is described in Section 2.2 ("Markov chain ensemble (MCE) method"). The possible values of the parameters are controlled by thresholds, which depend only on the external requirements to the precision of the MCE output. We discuss the sensitivity of the results to the parameter choice in an added Section 2.2.1 ("Parameter sensitivity").

We added the following text starting from line 136:

The parameter selection in the MCE method is constrained by the chosen thresholds and does not require manual tuning. The parameter thresholds are chosen based on the requirement for the method output precision. The purpose of this study is to

demonstrate the potential benefits of using Markov chains for constructing a weighted ensemble mean and we do not require high precision of the final evaluation metrics for our analysis and comparison with other methods. For other applications of this method, the appropriate thresholds should be chosen considering that higher precision requirements may increase computational load. We illustrate sensitivity of the MCE method to changes in $\sigma$ and $L$ on CMIP5 data in Figure 2.

The above illustration is a randomly chosen run of the MCE algorithm and resembles a typical behavior of its parameters. The cluster of R-squared values an the left side of the left graph represents the changes of $[\sigma_1, \sigma_2]$ interval at Step 9 in Table 1. As taking different values of $\sigma$ from the latest $[\sigma_1, \sigma_2]$ interval or increasing $L$ will not significantly change $R^2$ values, we use this approach for further analysis.

The paper also employs only a weak out of sample test, splitting the available data into a training and validation set. This is insufficient for the climate problem - where only the past is observable and the primary unknowns are climate projections in the future. Models with comparatively similar past trajectories might diverge significantly in the future - and the validation scheme considered here (where random timesteps are withheld) does not capture this divergence. A stronger test is a perfect model study, where individual models are withheld from the ensemble and late century projected performance is considered.

The perfect model test is covered in model-as-truth performance assessment as described above.

The paper considers that, for a given output variable, that the climate models should be weighted only by their fidelity in producing that variable - but there is no clear reason why this should be the case. Climate projections are functions of the integrated climate system which determine global climate sensitivities and regional feedbacks. As such, fidelity in producing a given variable (such as a regional temperature timeseries) in the past is no guarantee of accuracy in the future (Sanderson 2012). Lorenz et al (2018), for example, discusses at length the relative utility of different variable types for constraining future model evolution.

This is a really good point, and we agree. Although the MCE cannot predict the future behavior which didn't occur in the past, it avoids overfitting on the training set (at least if compared to such methods as COE). Similarly to the other ensemble methods, its prediction efficiency is naturally limited by ability of the input models to represent observations. As mentioned in the Section 2.2.2 ("MCE method limitations"), the uncertainty quantification including the analysis of different sources of error is too large to be included in the same paper together with the method introduction itself. This topic is therefore a subject for further MCE method development and is currently out of scope for this publication.

Finally, the paper does not acknowledge the various reasons climate models may disagree - natural variability provides an absolute limit on the performance of an uninitialized climate model, and so some discussion is required on how different sources of error would impact the results in this case.

Given these issues, I suggest the following revisions:

1 - outline more clearly (and ideally quantify) the benefits of using MCE beyond absolute skill scores (where the method will be outperformed by construction by Herger 2018)

Response:

We extended evaluation of the MCE method as discussed above and included the following text in the

Introduction (Section 1) starting from line 35:

It naturally produces non-negative weights that sum to one and captures some of the non-linear patterns in ensemble (here we refer to non-linear patterns as dependencies across time series, e.g. "if model A is the closest to the observations at time $t$ then model B is often the closest to the observations at time $t+1$"). It performs well on a range of data sets when compared to the standard simple mean and linear optimisation weighting methods as we demonstrate below. While it does not directly address the issue of model interdependence, it can mitigate some of the problems faced having dependent models in an ensemble.

Discussion (Section 5) starting from line 270:

The obtained results indicate that Markov Chain can be used to construct a better performing weighted ensemble mean with lower RMSE and higher R-squared values on validation data than commonly used methods like multi-model ensemble average and convex optimisation (Tables 2, 3 and 4). While it performed worse than COE on the training periods, we are confident that it is less prone to over-fitting than linear optimisation methods. We attribute this advantage of the MCE method to its ability to maintain the ensemble's diversity while optimising its weights on the training period (Figures 4, 5 and 6), to mitigate model interdependence and to capture some of the non-linear patterns in the data.

and in Conclusions (Section 6) starting from line 300:

The results of this study indicate that applying nonlinear ensemble weighting methods on climate data sets can improve future climate projection in terms of accuracy. Even a simple nonlinear structure such as Markov chain shows better performance on different commonly-used data sets than linear optimisation approaches. These results are supported by using standard performance metrics, cross-validation procedures and model-as-truth performance assessment. The developed MCE method is objective in terms of parameter selection, has a sound theoretical basis and has a relatively low number of limitations. It maintains ensemble diversity, mitigates model interdependence and captures some of the non-linear patterns in the data while optimizing ensemble weights. Based on the above, we are confident to suggest its application on other data sets and its usage for the future development of new nonlinear optimisation methods for weighting climate model ensembles.

2 - present the parameter sensitivity of the method in the main text

Response:

We have changed the method to eliminate the need for manual tuning and added Section 2.2.1 ("Parameter sensitivity").

3 - perform a rigorous leave-one-out test of weighting scheme performance in a perfect model projections of century-scale climate change

Response:

We performed a rigorous leave-one-out test on increased sample size in Section 4 ("Model-as-truth performance assessment").

4 - Consider how to address model interdependency in this method (which would bias the leave-one-out test)- either by adaptation of existing approaches or otherwise

Response:

We explained and illustrated how the MCE method is addressing model interdependency in Section 2.7 ("Model interdependence").

5 - Reconsider the relationship between variables used for weighting and those variables which are to be weighted (there is no reason to limit consideration in the weighting term to those variables which are themselves being weighted).

Response:

That would require a major redesign of the method and can be a topic for the future MCE development.

6 - consider the role of various sources of error and whether they should be represented within the scheme (natural variability, forcing errors, structural differences).

Response:

That would require a major restructuring of the publication with new data sets and according tests, discussions and conclusions as the topic is too large to include into the current manuscript.

Referee 2:

In this study, the authors present a novel method that employs Markov Chains as a means to weight members of global climate model (GCM) ensembles. Using three case studies involving historical simulations of global average temperature, regionally downscaled seasonal temperature, and a regional heat wave heuristic, they compare the performance of three model weighting schemes: simple model averaging (by definition, equal weights), a 'Convex Optimization Ensemble' (COE) method, and their 'Markov Chain Ensemble' (MCE) approach. Standard observational datasets from the recent past (up to ~120 years) are used for comparison based on RMSE and R2 skill scores. The proposed approach is interesting, and could be quite useful as a means to weight model ensembles and its simplicity is attractive, while also presenting a less ad-hoc approach than simple model averaging. However, I cannot evaluate the scientific merit of the approach because the model-observation tests are ill-posed in their current form.

The main problem is that comparing unfiltered GCM time series to observations is very problematic when applying typical skill scores because the interannual variability will not correspond between CMIP5 models and the observations. So although the underlying trend or evolving signal (assuming there is a signal, such as in global temperatures) of a perfectly performing model should match what is observed, the full time series from the model would not necessarily match the observed time series. This is because in the CMIP5 experiments, the GCMs begin the experiments with different initial conditions, and different model runs within the same model will begin at different points in the same control run before then beginning the perturbation experiment (i.e. including anthropogenic forcings). This makes direct time series comparisons very tricky if not handled carefully. Take a very simple example as shown in the example figure. Here are three simulated white noise time series (mean = 0, standard deviation = 1) overlaid onto two linear trends (0.1 for the black and blue time series, 0.03 for the red time series). The blue and red time series symbolize what could occur in a multimember CMIP5 ensemble. Note that even though the blue time series has exactly the same trend as the 'observed' time series, the RMSE is higher than the red time series simply because the inter-annual variation is misaligned (of opposite sign

in this case) with the observed anomalies. In contrast, the red time series has a lower RMSE, despite the fact that it does not capture the true forced trend. But the anomalies are aligned perfectly with observations. The red time series would be weighted higher in this case. It is 'right' for the 'wrong' reasons. Similarly the test set up in this manuscript is subject to the same problem. Individual models could exhibit anomalies that are more similar to the observed time series (driving down the skill scores), while the model response to the perturbation is less accurately simulated than other models with higher (by-chance) anomaly errors. Of course, all the methods that were tested (AVG, COE, MCE) could be similarly biased, in which case perhaps the results hold. But, there is no way to ascertain that in the current test structure.

Thank you for pointing out this common problem. As you mentioned this issue is not unique to MCE and is difficult to handle for all ensemble weighting methods. To address it in this publication we have now added additional model-as-truth (in- and out-of-sample) performance assessments in Section 4 ("Model-as-truth performance assessment"). The assessment is done on CMIP5 monthly data to increase the sample size and contains both performance (RMSE) and climatology (mean, trend) metrics. The text included in the updated manuscript can be found in the answer to the first reviewer on page 1.

This performance assessment helps to evaluate how sensitive the MCE method is to the problem described in your example. In this example both mean and trend would have larger errors in the long run if optimized only on RMSE (i.e. by giving a higher weight to the red line) compared to a simple averaging. We demonstrate that this is not the case on CMIP5 and NARCLiM monthly data as the projected mean and trend are improved compared to a simple averaging on both initial and extended data sets. Hence, we believe that the MCE method is robust against such issues and does not sub-optimize for RMSE only.

My other, more minor comments relate to similar issues with the structure of the model evaluation exercise. It is unsurprising that the model RMSE and R2 values were so poor when comparing GCM results to heat wave heuristics based on local weather station data (and again, where the model internal variability would have no reason except by chance to match the observed internal variability). The GCMs were developed at a scale that was never intended to resolve such localized patterns, and of course any annual heat waves that were observed, would only by chance occur in the same years (and be of similar magnitude) in the ensemble members.

Response:

We agree with the outlined challenges in constructing ensembles of RCMs. The main purpose of including those data sets was to demonstrate versatility of the MCE method due to its fewer limitations compared to many other optimization methods. We believe that we were able to demonstrate that the MCE method can be successfully applied on data with distributions that are very different from normal and with value constraints (non-negative in case of heatwaves). In fact, it shows higher performance improvements over simple averaging in terms of RMSE and $R^2$ metrics, as well climatology in more challenging scenarios like RCMs ensembles.

To improve the evaluation I suggest the authors revisit the literature to see how others have tackled this problem. More attention should be paid to, for example, efforts by Sanderson et al. (2015) to carefully construct valid comparisons between GCM ensembles and observations (in this case, by focusing the model skill evaluation on climatologies, rather than time series anomalies) while also taking into account model inter-dependence; something which the authors admit they do not account for in their method.

Others have addressed the non-initialized climate model/observation comparison problem by comparing long-term trends (e.g. Terando et al. 2012), which removes some of the problems with mis-matched internal variability, and gets closer to an actual forecast verification approach, but does not address other issues such as the robustness and reliability of weighting methods (see discussion in Knutti et al. 2017). It should be possible to construct a rigorous test of their MCE method, but the numerous challenges that have been widely and repeatedly documented in the literature should be acknowledged and addressed.

Response:

We analyzed both climatology and trend values in Section 4 ("Model-as-truth performance assessment") and increased the data sample size to have more confidence in the MCE performance and applicability. The analysis results support our initial statements regarding the potential benefits of using non-linear methods (in our study - Markov Chains as one of the most basic non-linear structures) for constructing a weighted ensemble mean.

Figures used in the revised manuscript:

[Figure]

**Figure 2.** Parameter sensitivity. Training (blue) and validation (red) R-squared values depending on $\sigma$ and $L$ values.

[Figure]

**Figure 3.** Change of model weights after adding highly correlated models with a low weight (Model 5) and a high weight (Model 8) as an additional ensemble member (Model 30). The initially chosen models are on the left in blue, the added Model 30 is on the left in red, the ensemble weights before (blue) and after(red) adding Model 30 are on the right.

[Figure]

**Figure 7.** Mean, trend and RMSE improvements (median, 25% and 75% percentiles) of $E_{MCE}$ compared to $E_{AVE}$ (CMIP5).

[Figure]

**Figure 8.** Mean, trend and RMSE improvements (median, 25% and 75% percentiles) of $E_{MCE}$ compared to $E_{AVE}$ (NARCLiM).

---

## Author Response (AR1)

**Response letter to the reviewers of "A Markov chain method for weighting climate model ensembles".**

Max Kulinich, Yanan Fan, Spiridon Penev, Jason P. Evan, Roman Olson

February 2021

**1 General**

We provide a point by point response to the referee's comments. Referee's original comments are in black fonts, and our response in blue font.

**2 Referee 1: Ben Sanderson sanderson@cerfacs.fr**

This study considers a novel application of Markov Chain methods to the problem of climate model weighting. The authors use a multi-model ensemble of climate model simulations, together with a range of different observation sources (global mean temperatures, regional averages and regional extreme temperature statistics). The novel "MCE" methodology is compared with two other approaches (a climatological weighted average score - Lambert and Boer 2001, AVE hereafter, and the ensemble transformation approach of Bishop and Abramowitz 2013, COE hereafter). The authors find desirable properties of their proposed approach in terms of out-of-sample skill and performance in non normally distributed quantities related to temperature extremes.

The approach shows promise, the potential for more robust error estimates in weights would be useful and the stochastic nature of the computation may offer additional benefits. However, these avenues are not explored in the current text and the authors have not yet fully addressed the basic requirements of an operational climate model weighting scheme (robust out of sample testing), and existing simpler approaches may outperform the approach presented here for the metrics considered (see below).

Response: Thank you for this suggestion, and we have now added additional model-as-truth (in- and out-of-sample) performance assessments in Section 3 ("Results"). The assessments are done on CMIP5 and NARCLiM monthly data using trend bias, climatology monthly bias, interannual variability and climatological monthly RMSE metrics.

We describe those metrics in sections 2.5.2 - 2.5.5, and the model-as-truth performance assessment procedure in section 2.6.2. We present the models-as-truth experiment results for each data set with monthly data (lines 242 - 248 and 262 - 266) followed by a discussion starting from line 284 and conclusions starting from line 305.

The introduction talks of the need to represent and consider model interdependency - but the actual study does not propose any mechanism for accounting for model interdependencies or common

components in the weighting scheme. Example approaches for doing so are laid out in Sanderson (2017) and Lorenz (2018).

We added a new section 2.2.2 "Model interdependence" to demonstrate how MCE is accounting for model interdependencies.

The study also generalizes existing literature by the two comparative cases considered (AVE and COE) as "linear" approaches, but there are more complex schemes in the literature which address issues not considered here, like optimal subselection (Sanderson 2015). In addition, given an RMSE or R^2 metric - methods which determine the global optimization of weighted scores (Herger 2018) are in theory unbeatable - so, although there may be tangential benefits to using MCE, it's unclear that the approach here could outperform a global weight optimization for these types of metric.

The size limit of the publication does not allow to compare the MCE method to all the existing ensemble weighting techniques, though it would be a valuable addition to the study results. Though the MCE technique can be easily outperformed on training data by other methods (as shown in Tables 4,5 and 6), when it comes to cross-validation the MCE performs generally well on validation data (on par or better than other methods). We attribute its high performance to its ability to capture some of the sequential information in the input time series, which other methods do not use. An example of such information would be: if model A is the closest to the observation at time t, then model B will be the closest at time t+1 with high probability. This would be naturally captured by the MCE method through the transition matrix P, but difficult (or impossible) to capture by linear or subselection methods. In addition, the MCE method maintains ensemble diversity on all three data sets and its performance does not degrade from training to validation as much COE.

The paper attributes differences in weighting behavior between MCE, AVE and COE to differences in methodology, without sampling the subjective degrees of freedom in each of the approaches. As such, the observation that the MCE approach tends to rule out fewer models than COE is potentially a function of the chosen L and sigma parameters as well as the MCE approach itself. Any revision should present parameter sensitivities in the main text.

We thank the referee for the above suggestion. We have now modified the MCE method to randomly select sigma from a segment limited by the exponential function properties. In addition the modified algorithm selects only one set of weights and is less sensitive to the number of simulations (L). The modified algorithm is described in Section 2.2 ("Markov chain ensemble (MCE) method"). We also discuss the effect of the parameter choice in section 2.2.1 ("Parameter sensitivity").

The paper also employs only a weak out of sample test, splitting the available data into a training and validation set. This is insufficient for the climate problem - where only the past is observable and the primary unknowns are climate projections in the future. Models with comparatively similar past trajectories might diverge significantly in the future - and the validation scheme considered here (where random timesteps are withheld) does not capture this divergence. A stronger test is a perfect model study, where individual models are withheld from the ensemble and late century projected performance is considered.

The perfect model test is covered in model-as-truth performance assessment as described above.

The paper considers that, for a given output variable, that the climate models should be weighted only by their fidelity in producing that variable - but there is no clear reason why this should be

the case. Climate projections are functions of the integrated climate system which determine global climate sensitivities and regional feedbacks. As such, fidelity in producing a given variable (such as a regional temperature timeseries) in the past is no guarantee of accuracy in the future (Sanderson 2012). Lorenz et al (2018), for example, discusses at length the relative utility of different variable types for constraining future model evolution.

This is a really good point, and we agree. Though the MCE cannot predict the future behavior which didn't occur in the past, it avoids overfitting on the training set (at least if compared to such methods as COE). As other ensemble methods its prediction efficiency is naturally limited by ability of the input models to represent observations. As mentioned in the Section 2.2.3 ("MCE method limitations"), the uncertainty quantification including the analysis of different sources of error is too large to be included in the same paper as the method introduction itself. This topic is therefore a subject for further MCE method development and is currently out of scope for this publication.

Finally, the paper does not acknowledge the various reasons climate models may disagree - natural variability provides an absolute limit on the performance of an uninitialized climate model, and so some discussion is required on how different sources of error would impact the results in this case.

Given these issues, I suggest the following revisions: 1 - outline more clearly (and ideally quantify) the benefits of using MCE beyond absolute skill scores (where the method will be outperformed by construction by Herger 2018)

Response: We extended evaluation of the MCE method as discussed above and included the following text in the Discussion (Section 5) starting from line 278:

The obtained results indicate that Markov chains can be used to construct a better performing weighted ensemble mean with lower RMSE on validation data than commonly used methods like multi-model ensemble average and convex optimisation (Tables 3, 5 and 7). As the method's performance did not degrade from training to validation as much as COE, we are confident that it is less prone to over-fitting than linear optimisation methods. We attribute this advantage of the MCE method to its ability to maintain the ensemble's diversity while optimising its weights on the training period (Figures 3, 5 and 7), to mitigate model interdependence and to capture some of the nonlinear patterns in the data.

The MCE method also performs at the same level as other methods in terms of climatological metrics and model-as-truth performance assessment, which gives us confidence in its ability to be used for future estimation of climate variables.

and in Conclusions (Section 6) starting from line 312: The developed MCE method is objective in terms of parameter selection, has a sound theoretical basis and has a relatively low number of limitations. It maintains ensemble diversity, mitigates model interdependence and captures some of the non-linear patterns in the data while optimizing ensemble weights. It is also shown to perform well on non-Gaussian data sets. Based on the above, we are confident to suggest its application on other data sets and its usage for the future development of new nonlinear optimisation methods for weighting climate model ensembles.

2 - present the parameter sensitivity of the method in the main text Response:

We have changed the method to eliminate need for manual tuning and added Section 2.2.1 ("Parameter sensitivity").

3 - perform a rigorous leave-one-out test of weighting scheme performance in a perfect model projections of century-scale climate change Response:

We performed a rigorous leave-one-out test as described in section 2.6.2 ("Model-as-truth performance assessment").

4 - Consider how to address model interdependency in this method (which would bias the leave-one-out test)- either by adaptation of existing approaches or otherwise

Response: We explained and illustrated how the MCE method is addressing model interdependency in Section 2.2.2 ("Model interdependence").

5 - Reconsider the relationship between variables used for weighting and those variables which are to be weighted (there is no reason to limit consideration in the weighting term to those variables which are themselves being weighted).

Response: That would require a major redesign of the method and can be a topic for the future MCE development.

6 - consider the role of various sources of error and whether they should be represented within the scheme (natural variability, forcing errors, structural differences).

Response: That would require a major restructuring of the publication with new data sets and according tests, discussions and conclusions as the topic is too large to include into the current manuscript.

**3 Referee 2**

In this study, the authors present a novel method that employs Markov Chains as a means to weight members of global climate model (GCM) ensembles. Using three case studies involving historical simulations of global average temperature, regionally downscaled seasonal temperature, and a regional heat wave heuristic, they compare the performance of three model weighting schemes: simple model averaging (by definition, equal weights), a 'Convex Optimization Ensemble' (COE) method, and their 'Markov Chain Ensemble' (MCE) approach. Standard observational datasets from the recent past (up to 120 years) are used for comparison based on RMSE and R2 skill scores. The proposed approach is interesting, and could be quite useful as a means to weight model ensembles and its simplicity is attractive, while also presenting a less ad-hoc approach than simple model averaging. However, I cannot evaluate the scientific merit of the approach because the model-observation tests are ill-posed in their current form.

The main problem is that comparing unfiltered GCM time series to observations is very problematic when applying typical skill scores because the interannual variability will not correspond between CMIP5 models and the observations. So although the underlying trend or evolving signal (assuming there is a signal, such as in global temperatures) of a perfectly performing model should match what is observed, the full time series from the model would not necessarily match the observed time series. This is because in the CMIP5 experiments, the GCMs begin the experiments with different initial conditions, and different model runs within the same model will begin at different points in the same control run before then beginning the perturbation experiment (i.e. including anthropogenic forcings). This makes direct time series comparisons very tricky if not handled carefully. Take a very simple example as shown in the example figure. Here are three simulated white noise time series (mean = 0, standard deviation = 1) overlaid onto two linear trends (0.1 for the

black and blue time series, 0.03 for the red time series). The blue and red time series symbolize what could occur in a multimember CMIP5 ensemble. Note that even though the blue time series has exactly the same trend as the 'observed' time series, the RMSE is higher than the red time series simply because the inter-annual variation is misaligned (of opposite sign in this case) with the observed anomalies. In contrast, the red time series has a lower RMSE, despite the fact that it does not capture the true forced trend. But the anomalies are aligned perfectly with observations. The red time series would be weighted higher in this case. It is 'right' for the 'wrong' reasons. Similarly the test set up in this manuscript is subject to the same problem. Individual models could exhibit anomalies that are more similar to the observed time series (driving down the skill scores), while the model response to the perturbation is less accurately simulated than other models with higher (by-chance) anomaly errors. Of course, all the methods that were tested (AVG, COE, MCE) could be similarly biased, in which case perhaps the results hold. But, there is no way to ascertain that in the current test structure.

Thank you for pointing out this common problem. As mentioned this issue is not unique to MCE and is difficult to handle for all ensemble weighting methods. To address it in this publication we have now added additional model-as-truth (in- and out-of-sample) performance assessments as described in section 2.6.2 ("Model-as-truth performance assessment"). The assessments are done on CMIP5 and NARCLiM monthly data using trend bias, climatology monthly bias, interannual variability and climatological monthly RMSE metrics. This performance assessment helps to evaluate how sensitive is the MCE method to the problem described in your example. In this example both mean and trend would have larger errors in the long run if optimized only on RMSE (i.e. by giving a higher weight to the red line) compared to a simple averaging. We demonstrate that this is not the case on CMIP5 and NARCLiM monthly where MCE is performing at the same level as AVE and COE. Hence we believe that the MCE method is at least as robust against such issues as the methods mentioned and does not sub-optimize to RMSE only.

My other, more minor comments relate to similar issues with the structure of the model evaluation exercise. It is unsurprising that the model RMSE and R2 values were so poor when comparing GCM results to heat wave heuristics based on local weather station data (and again, where the model internal variability would have no reason except by chance to match the observed internal variability). The GCMs were developed at a scale that was never intended to resolve such localized patterns, and of course any annual heat waves that were observed, would only by chance occur in the same years (and be of similar magnitude) in the ensemble members.

Response: We agree with the outlined challenges in constructing ensembles of RCMs. The main purpose of including those data sets was to demonstrate versatility of the MCE method due to its fewer limitations than many other optimization methods. We believe that we were able to demonstrate that the MCE method can be successfully applied on data with distributions that are very different from normal and with value constraints (non-negative in case of heatwaves).

To improve the evaluation I suggest the authors revisit the literature to see how others have tackled this problem. More attention should be paid to, for example, efforts by Sanderson et al. (2015) to carefully construct valid comparisons between GCM ensembles and observations (in this case, by focusing the model skill evaluation on climatologies, rather than time series anomalies) while also taking into account model inter-dependence; something which the authors admit they do not account for in their method. Others have addressed the non-initialized climate model/observation comparison problem by comparing long-term trends (e.g. Terando et al. 2012), which removes some

of the problems with mis-matched internal variability, and gets closer to an actual forecast verification approach, but does not address other issues such as the robustness and reliability of weighting methods (see discussion in Knutti et al. 2017). It should be possible to construct a rigorous test of their MCE method, but the numerous challenges that have been widely and repeatedly documented in the literature should be acknowledged and addressed.

Response: We analyzed climatologies using trend bias, climatology monthly bias, interannual variability and climatological monthly RMSE metrics with both cross-validation (holdout method) and model-as-truth performance assessment to have more confidence in the MCE performance and applicability.

---

## Author Response (AR2)

**Response letter to the reviewers of "A Markov chain method for weighting climate model ensembles".**

Max Kulinich, Yanan Fan, Spiridon Penev, Jason P. Evan, Roman Olson

April 2021

**1   General**

Referees' original comments are in black font in Section 1 and Section 2, and our response is in blue font in Section 3.

**2   Referee 1: Ben Sanderson sanderson@cerfacs.fr**

Thanks to the authors for the detailed response, and for making efforts to address some of the concerns from the first submission. The addition of the leave-one-out tests, interdependency testing and parameter sensitivity study add robustness to the paper - and I'm now technically happy to see it published.

However, there are some points made in the original review which still require a caveat in the discussion. The sole consideration of globally integrated time-series as the outputs of a model ignores a lot of potentially useful data, and ultimately limits this analysis. For example - two models may have very similar global mean warming timeseries for the 20th Century, but with very different physical representations and regional climates. These two models should not, in practice, be considered to be highly related - but the method listed here would consider them to be so. The integrated response is a very low dimensional metric, and is unlikely to be informative about shared model assumptions. Other studies (Masson et al 2011, Sanderson 2017) have found that it is primarily complex spatial fields which are most informative on actual shared model components. I appreciate that implementing further analysis is out of scope, but this issue is a caveat to the current approach, and should be noted for consideration in future study.

Sanderson, B. M., Wehner, M., Knutti, R. (2017). Skill and independence weighting for multi-model assessments. Geoscientific Model Development, 10(6), 2379-2395.

Masson, David, and Reto Knutti. "Climate model genealogy." Geophysical Research Letters 38, no. 8 (2011).

**3   Referee 2: Anonymous Referee 2**

The paper is definitely improved. In particular, the cross-validation tests are a good and needed addition. And in terms of the comments I raised, the additional performance metrics listed in Section 2.5 (Trend Bias, Climatology monthly bias, Interannual variability, and Climatological monthly

RMSE) help to provide a much more salient and defensible measure of the method's performance in this context, at least as applied to the CMIP and NARCLiM data. With respect to the KMA case study, where the expanded set of performance metrics is not evaluated (since there is no seasonality in the heuristic), I understand that the idea is to show how the method performs in non-Gaussian or data-censored situations like the heat stress choice. I also understand that it's the same model ensemble being tested among the three methods, so it's a "fair fight" regardless of the physical meaning (or not) of the respective weights. But I still have a problem with presenting results from an ill-posed test without stating that these do not represent weights with real-world interpretability. So in reference to Section 3.3, there should be text added somewhere explicitly stating that a priori we would not expect these RMSE values to have any physical interpretation related to model skill. Or a short caveats section could be added to address these points. So while it's fine to use these case studies as toy problems to test the method's performance in different statistical contexts, the authors need to be clear that that's what this is.

**4    Response to referees' comments**

Thank you for your feedback and improvement suggestions. We acknowledge the remarks by both referees, that the physical interpretability of the weights is limited and can be improved in the future work by applying the MCE method on spatially distributed data.

We have now added the caveats in Section 2.1 (Data), lines 88-89:

"In this pilot study we use spatially averaged data, which limits physical interpretability of the model weights, but the method can be extended to spatially distributed data."

and in Section 2.2.3 (MCE method limitations), lines 173-174:

"Finally, as the MCE method does not consider spatial information, the resulting weights have limited physical interpretability. Extending the MCE method to utilize such information is a subject for future research."

and in Section 4 (Discussion), lines 291-293:

"However, as previous studies show (e.g. Masson and Knutti (2011); Sanderson et al. (2017)) and as discussed in Section 2.2.3, extending the MCE method to include spatial information would improve our ability to interpret the physical meaning of the resulting weights."